# Fresh Air for the Mire-Breathing Hypothesis: *Sphagnum* Moss and Peat Structure Regulate the Response of CO₂ Exchange to Altered Hydrology in a Northern Peatland Ecosystem

Ally O'Neill [1] , Colin Tucker [2,*] and Evan S. Kane [1,2]

1    College of Forest Resources and Environmental Sciences, Michigan Technological University, Houghton, MI 49931, USA
2    USDA Forest Service Northern Research Station, Houghton, MI 49931, USA
*    Correspondence: colin.tucker@usda.gov; Tel.: +1-307-4608303

**Abstract:** *Sphagnum*-dominated peatlands store more carbon than all of Earth's forests, playing a large role in the balance of carbon dioxide. However, these carbon sinks face an uncertain future as the changing climate is likely to cause water stress, potentially reducing *Sphagnum* productivity and transitioning peatlands to carbon sources. A mesocosm experiment was performed on thirty-two peat cores collected from two peatland landforms: elevated mounds (hummocks) and lower, flat areas of the peatland (hollows). Both rainfall treatments and water tables were manipulated, and CO₂ fluxes were measured. Other studies have observed peat subsiding and tracking the water table downward when experiencing water stress, thought to be a self-preservation technique termed 'Mire-breathing'. However, we found that hummocks tended to compress inwards, rather than subsiding towards the lowered water table as significantly as hollows. Lower peat height was linearly associated with reduced gross primary production (GPP) in response to lowered water tables, indicating that peat subsidence did not significantly enhance the resistance of GPP to drought. Conversely, *Sphagnum* peat compression was found to stabilize GPP, indicating that this mechanism of resilience to drought may transmit across the landscape depending on which *Sphagnum* landform types are dominant. This study draws direct connections between *Sphagnum* traits and peatland hydrology and carbon cycling.

**Keywords:** carbon cycle; Histosol; peat; climate change; *Sphagnum*; soil respiration; subsidence





## 1. Introduction

Northern peatlands are globally important carbon sinks, storing more carbon than all of Earth's forests [1–4] despite taking up only 3% of the global land area [5]. However, the carbon sink status of peatlands is vulnerable because climate warming and land conversion are currently inducing a suite of changes to vegetation and hydrology that will impact peat formation and storage [1]. For instance, as the climate continues to change, it is likely that warmer temperatures will cause drops in water table (WT) position [6]. When water tables are lowered, the oxic zone of peat is expanded, causing increases in decomposition rates [7]. At the same time, water stress caused by lowered water tables is likely to reduce rates of photosynthesis, therefore reducing net primary production (NPP) [8,9]. If decomposition exceeds production, peatlands will no longer function as carbon sinks. Similarly, large-scale ditching to convert peatlands to agriculture and forestry has caused substantial regional losses of peat [10].

*Sphagnum* moss is a key player in the formation of peatland ecosystems, and alone is responsible for the storage of around 10% of the world's soil carbon [11]—more than any other plant genus [12]. *Sphagnum* lacks a vascular system, and relies solely on the capillary transport of water to survive, resulting in potentially high susceptibility to water stress [12,13] such as more frequent or prolonged droughts that are predicted to occur with climate change [7,14,15]. Many northern peatlands contain a diverse microtopography

formed by *Sphagnum* growing in hummocks and hollows [16], defined primarily by their relative heights with an alternating hummock-hollow microtopographical pattern [17]. Because they grow further above the water table than hollow mosses, hummock mosses have adaptations that allow them to tolerate (and even create) drier conditions than hollow mosses can tolerate [18,19]. For instance, capitula in hummocks tend to be smaller and denser than those in hollows as more resources are put towards stem growth which in turn increases the efficacy of capillary transport and retention of water from deeper in the peat [11,19,20]. For this reason, we hypothesized that hummocks would be more resistant to drought and lowered water tables than hollows, maintaining more stable $CO_2$ cycles.

It has been observed that *Sphagnum* dominated peatlands often undergo changes in surface height as a result of fluctuating water tables [21–24]. This following of the water table is often referred to as "mire-breathing" and is thought to be a method of self-preservation under water stress [25], which has been documented at scales of entire peatlands, with variations related to the height above water table of different peatland landforms [26]. In the traditional model, the collective peat surface expands as the water table position rises, raising the bog surface and increasing water storage. With a decline in water table position the peat loses volume, and the peat surface drops, which maintains close contact between *Sphagnum* mosses and the water table—thereby maintaining their productivity. While prior work has shown there to be different water transport and storage characteristics among different peatland microforms [27], it is unknown whether the contrasting *Sphagnum* landforms (i.e., hummocks and hollows) exhibit similar mechanisms of mire-breathing and how that affects their ability to maintain productivity under drier conditions.

Despite the critical influence *Sphagnum* dominated peatlands have in the global carbon cycle, ecosystem and climate models are only beginning to take *Sphagnum* moss function in peatlands into account [1,3,28–30]. Additionally, data regarding the role *Sphagnum* architecture may play in stabilizing peatland carbon storage as the climate warms is lacking [24]. This study aims to identify whether differing *Sphagnum* types are more resistant to water stress than others, and how carbon cycling may be impacted by that resistance. Although it is well known that some physical traits of *Sphagnum* may increase moss resistance to water stress, to our knowledge the relationship between these traits and the possible stabilizing factors that they may have on carbon cycling is not well understood. We conducted this study with the goal to better understand the fate of carbon storage and emissions on both the species and landscape scale as *Sphagnum* dominated peatlands face imminent water stress.

## 2. Materials and Methods

### 2.1. Peat Harvest and Water Table Treatments

Thirty-two cores of peat (~30 cm depth × 20 cm diameter) were collected from a *Sphagnum*-dominated oligotrophic peatland in Nestoria, MI (46°34′22.66″ N, 088°16′44.85″ W) previously described by [31]. Briefly, the overstory consisted of *Larix laricina* and *Picea mariana* and the understory consisted of mixed sedge and ericoid shrubs. Peat cores were obtained using a sharpened PVC corer and serrated knives. Cores were collected from two peatland landforms: elevated mounds (hummocks) and lower, flat microforms (hollows/lawns, hereafter 'hollows') in a relatively open, homogeneous area within the peatland. Each peat core was placed in two 3.8 L plastic bags left open at the top and arranged in 35 cm tall × 20 cm diameter PVC tubes. The full dynamic range of "mire breathing" for hummock/hollow microtopography can range between 20–60 cm depth [32], with the majority of changes in water storativity occurring in the top 40 cm [27]. As such, the top 30 cm measured in this study is likely reflective of this range but does not necessarily capture dynamics occurring within denser peat. Cores were then transported to the Houghton Mesocosm Facility at the USDA Forest Service Northern Research Station, where they were subjected to experimental treatments modeled after a larger mesocosm experiment [33]. Prior to initiating treatments, to allow the samples to acclimate to the new environment, each core was kept saturated for approximately two weeks in the outdoor

mesocosm facility. All vascular plants were then clipped, leaving only *Sphagnum* mosses and a low cover (<5% area) of *Polytrichum strictum* moss. Each sample (hereafter referred to as 'sphagnocosm') was randomly assigned a treatment of either a high or low water table depth coupled with an average or droughted rainfall. The experiment was thus a fully crossed design with water table depth × rainfall × landform, with 4 replicates per treatment level (Figure 1).

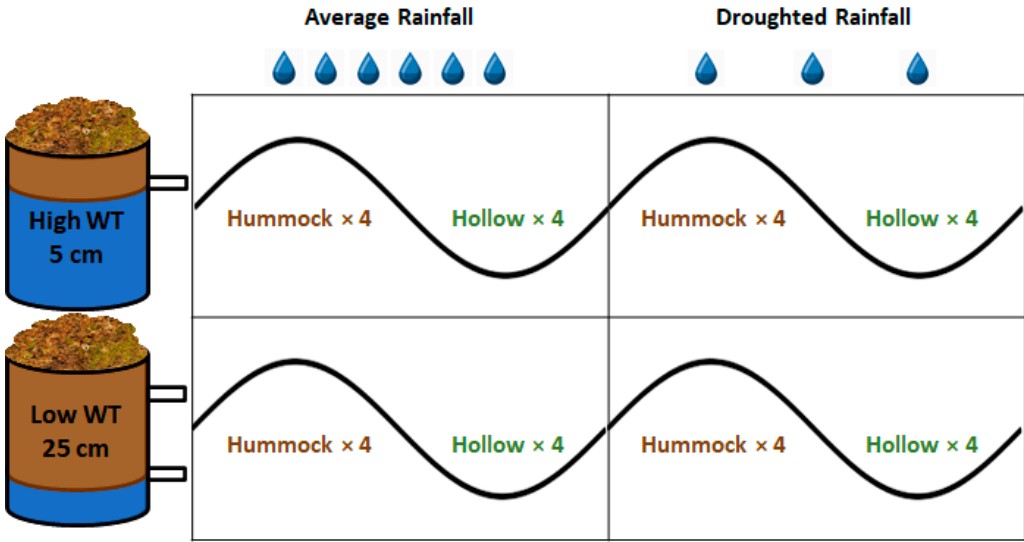

**Figure 1.** The experimental design was a fully factorial combination of landform (Hummock vs. Hollow), water table (High vs. Low), and Rainfall (Average vs. Droughted), with 4 replicate sphagnocosms per treatment level. Each sphagnocosm received an average (400 mL) or droughted (200 mL) rainfall treatment, along with a high (5cm from the initial peat surface) or low (25 cm from the initial peat surface) water table treatment, which were maintained via drainage valves indicated on the left of the figure.

Water table treatments were implemented by first measuring the average surface height of the *Sphagnum* in each bin. A collar with a wire grid was placed on each bin, and a ruler was dropped gently from four points formed by the grid until it rested on the surface of the moss. The point where the ruler lined up with the top of the collar was recorded (cm), and the height of the collar was then subtracted from the recorded height. One drainage hole was drilled in each sphagnocosm 5 cm below the average surface height for that sample, and for the low water table treatment an additional hole was drilled at 25 cm below the average surface height. Water was allowed to drain freely from each hole, and small tubes were inserted in each hole to help with drainage. Rainfall treatments were initiated at the same time as water table depth treatments. Sphagnocosms were kept under clear plastic rainout shelters to ensure that all inputs of water were accounted for. Half of the sphagnocosms received a rainfall treatment representing an average monthly summer rainfall of 163 mm while half received a droughted rainfall reduced by ~50% from the average: converting the average rainfall this to units of mL/bin and then dividing by the number of rainfall events resulted in a volume of 400 mL/rain event in the "average" rainfall treatment, and 200 mL/rain event in the "droughted" rain event. Sphagnocosms were watered three times a week over the 25 day duration of the experiment using artificial rainwater (See Potvin et al. 2015 [33] for rainfall chemistry).

### 2.2. Flux Measurements

$CO_2$ fluxes were measured twice a week over four weeks using a mobile cavity ring down spectrometer (Picarro GasScouter G4301; Picarro, Inc. Santa Clara, CA, USA), and a clear acrylic chamber (6280 cm$^2$) that fit tightly over each PVC tube. Flux measurements were taken between 9 am and 3 pm each day to ensure maximum sunlight. Measurements

were taken over the course of two minutes first in full sunlight (measuring Net Ecosystem Exchange (NEE) and then under darkness (measuring ecosystem respiration (R), using an opaque cloth to block out all sunlight. Gross primary production was determined as the difference between R and NEE. PAR was recorded for each light measurement, the air temperature was recorded before and after each light and dark measurement, and the temperature inside the bin was also recorded at the end of each measurement. The order flux measurements were taken was randomized each measurement cycle. Over the course of a summer, 640 flux measurements were recorded in total.

### 2.3. Spectral Reflectance Measurements

Spectral indices closely related to *Sphagnum* primary productivity and water content were calculated and compared to flux measurement data [31,34–36]. Changes in spectral water content (wetness index WI, floating water band index FWBI) and photosynthetic capacity (chlorophyll index CI, normalized difference vegetation index NDVI) were measured using an ASD Fieldspec 3 spectroradiometer (Analytical Spectral Devices, Boulder, CO, USA) under clear, low haze conditions as previously described in detail [31,36]. The spectroradiometer was held 30 cm above the surface of the center of each sphagnocosm, and was white referenced every ten minutes to adjust for changes in sun angle and sky conditions.

### 2.4. Sphagnum Characteristics

Species surveys of the sphagnocosms were conducted using the point-intercept method [33,37]. A grid with 25 pre-defined squares was placed on top of each sphagnocosm, and a thin rod was gently dropped at the bottom corner of each square. The species closest to or touching the rod was recorded. *S. fallax* was found to dominate the hollow/lawns while *S. fuscum* dominated the hummock samples. *S. magellanicum* and *S. angustifolium* and *Polytrichum strictum* were also present but rarer [31]. To measure capitula density per species in each sphagnocosm, we counted each *Sphagnum* capitulum within two 4 cm squares in predetermined locations in each sample. Moss height measurements of each sphagnocosm were taken once a week for five weeks using the method described in the treatment section. Spectral measurements were taken on 6 and 7 August 2020 as described above, with a nominal spectral range spanning 325–1075 nm [36].

### 2.5. Data Analysis

The response of $CO_2$ fluxes to water table and rainfall treatments, and landform, were analyzed using linear mixed effects modeling, to account for repeated measurements on the same mesocosm over time. The effect of water table and rainfall treatments, and landform, on surface height and capitula density was evaluated via analysis of variance. The relationship between surface height and capitula density was evaluated using correlation analysis. Finally, the full suite of effects was evaluated by structural equation modeling. Data were analyzed using 'R' software version 4.03 (Vienna, Austria) [38], and packages 'nlme' [39] and 'lavaan' [40].

## 3. Results

### 3.1. CO$_2$ Fluxes in Response to Treatments

Net ecosystem exchange (NEE) increased significantly with lowered water tables (Figure 2; $p$-value $\leq 0.001$; see Supplementary File S1 for time series of $CO_2$ fluxes). *Sphagnum* type was a determining factor in NEE response to lowered water tables, as NEE in hollows was more greatly impacted by the low water table treatment than NEE in hummocks ($p$-value = 0.0086), and water table depth contributed significantly to the difference in NEE between hummocks and hollows ($p$-value $\leq 0.001$). Rainfall had no significant impact on NEE in either *Sphagnum* type ($p$-value = 0.4398). Water table depth ($p$-value = 0.2955), *Sphagnum* type ($p$-value = 0.4309), and rainfall ($p$-value = 0.5396) had no significant effects on respiration. However, low water tables coupled with droughted rainfall had a marginally significant effect on respiration ($p$-value = 0.0574). Gross primary production

(GPP) was significantly decreased by low water table depths (*p*-value = 0.0227). Although, hollow GPP decreased more dramatically as a result of lowered water tables than the GPP of hummocks (*p*-value = 0.0452). Rainfall alone had no significant effect on GPP (*p*-value = 0.7005). However, when coupled with low water tables, droughted rainfall did reduce GPP (*p*-value = 0.0452).

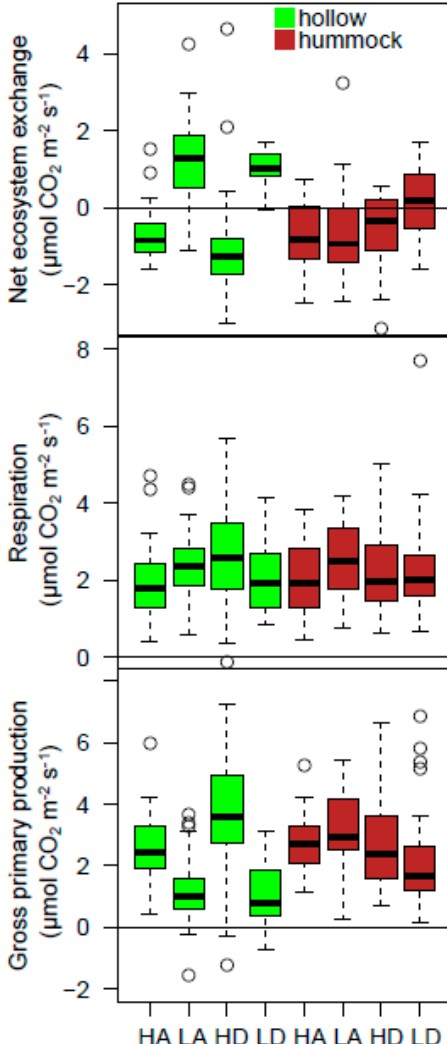

**Figure 2.** Boxplots of fluxes in response to experimental treatments. The treatment labels are "H" = high water tables, "L" = low water tables, "A" = average rainfall, "D" = droughted rainfall. Green boxes are hollows, brown boxes are hummocks. Box intervals represent the first and third interquartile range, whiskers represent the minimum and maximum after removing outliers, and the horizontal bold line represents the median.

### 3.2. Sphagnum and Peat Physical Structure in Response to Treatments

Capitula density was most significantly affected by whether moss was a hummock or hollow (*p*-value= $7.69 \times 10^{-5}$). Capitula density in hummocks exposed to low water tables was significantly greater than hummocks exposed to high water tables (*p*-value adj = 0.0373) and capitula density in hollows across both WT treatment levels (*p*-value adj = 0.0001). There was no significant difference in capitula density between hummocks and hollows treated with high water tables (*p*-value adj = 0.5128). Hollow capitula density did not show any significant difference between high and low water table treatments (*p*-value adj = 0.9794). Increased capitula density was associated with higher resistance of GPP to lowered water tables, such that this resistance was higher in hummocks compared to hollows.

Water table depth was found to have the greatest effect on peat surface height ($p$-value = $3.70 \times 10^{-7}$, see Supplementary File S1 for time series of the change in peat surface height). Whether a bin was a hummock or hollow did not significantly affect surface height ($p$-value = 0.09193). However, the interaction between *Sphagnum* type and water table depth did significantly impact surface heights ($p$-value = 0.00304). A mean significant difference of 3.57 cm in surface height was recorded between hollows subjected to low water table treatments and hollows subjected to high water table treatments ($p$-value = $7 \times 10^{-7}$). A mean difference of only 1.18 cm was recorded between hummocks with high water table treatments and hummocks with low water table treatments ($p$-value = 0.134483). It was observed that hummocks tended to compress inwards, rather than subsiding towards the lowered water table as significantly as hollows (Figure 3). Lower peat height was linearly associated with reduced GPP in response to lowered water tables, indicating that peat subsidence did not significantly enhance the resistance of GPP to lower water tables. Conversely, *Sphagnum*/peat inward compression was found to stabilize GPP under lowered water tables (Figure 3).

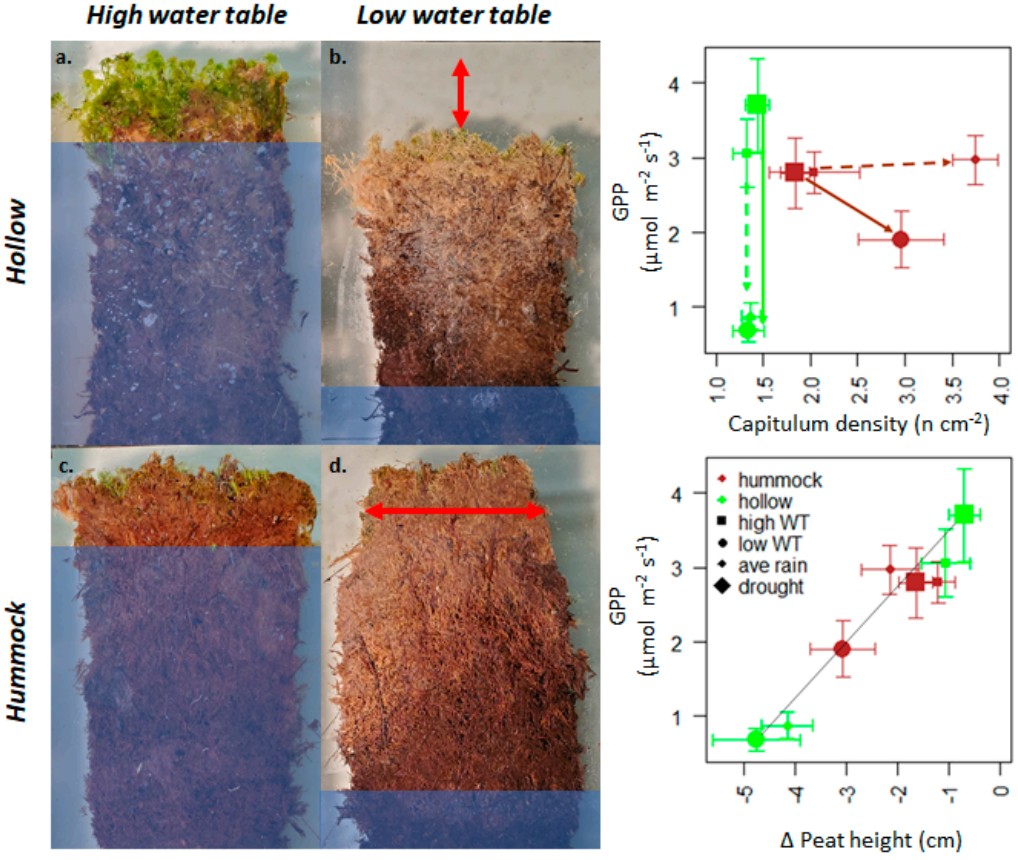

**Figure 3.** The relationship between water tables, peat structure and GPP. The images show Sphagnocosm monoliths removed from the PVC collars and cut into cross sections, representing hollows (**a**,**b**) and hummocks (**c**,**d**), under high (**left**) and low (**right**) water tables. The shape of the monolith was reconstructed based on the measured decrease in peat height and increased density (inward compression) of capitula measured in situ, the pictured change shows the mean decrease in peat height or increased density from a high water table—average rainfall treatment to a low water table drought treatment, with the red arrows representing the primary axis of change in each image. The figures on the right show GPP in relation to e. capitula density and f. surface subsidence (Δ peat height). Error bars represent standard error. In the upper right graph, dashed arrows connect the the high and low water table treatments at average rainfall, while solid arrows connect the high and low water table treatments at drought, with colors representing the respective landforms. The line in the lower right figure represents significant linear fit across treatments.

## 4. Discussion

### 4.1. Sphagnum Trait Mechanisms and Consequences

In this study, we investigated the changing carbon dioxide cycles of different *Sphagnum* landform types subjected to water stress, and furthered our insight of peatland carbon and hydrological dynamics. We found strong support that the carbon cycles of hummock landforms are more resilient to water stress than hollows (Figure 2). This indicates that denser *Sphagnum* moss in peatlands does increase resilience to drought. Our findings also indicate that denser capitula—such as those found in hummock species—are associated with increased gross primary production (Figures 2 and 4; Table 1). Additionally, under low water tables, increased capitula density does seem to increase the resilience of $CO_2$ cycles (Figure 2). Our data suggests that carbon cycling of northern peatlands may be affected on a landscape scale, depending on which *Sphagnum* landform types are dominant (Figure 4; Table 1). Peatlands with more hummock landforms may respond better to climate warming and water stress as carbon fluxes remain more stable, while hollow dominated peatlands may degrade and become carbon sources at a faster rate. However, we did find that hummock sphagnocosms became carbon sources (NEE became positive) when lowered water tables and droughted rainfall were combined. The differing responses of *Sphagnum* landform types to our treatments and the ability of some moss traits—such as capitula density—to stabilize carbon cycles reveal that *Sphagnum* and *Sphagnum* dominated peatlands are not static but are rather responsive ecosystems with a great amount of plasticity.

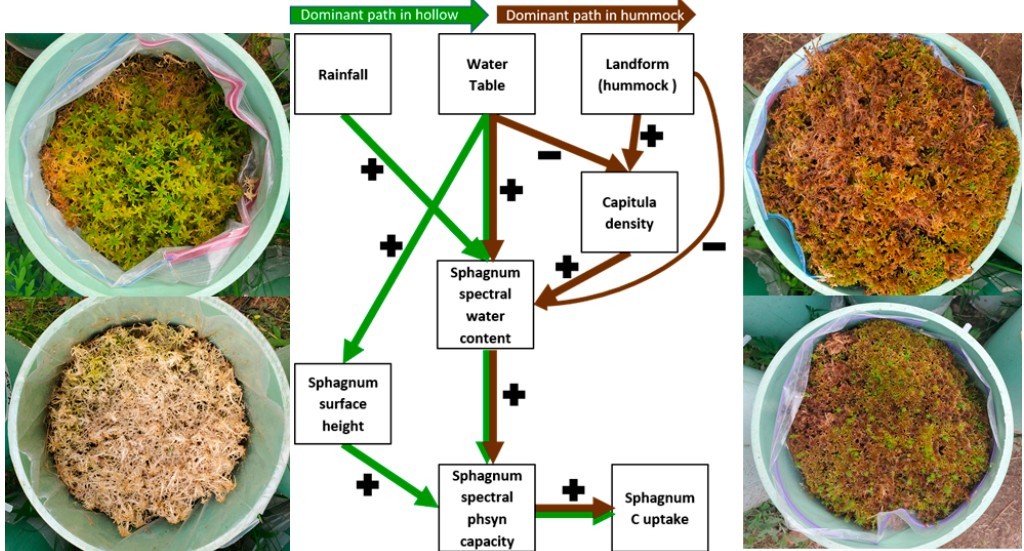

**Figure 4.** Structural equation model of *Sphagnum* C uptake (GPP) as a function of the treatments, changes in physical structure, and spectral indicators of *Sphagnum* spectral water content (Wetness Index) and spectral photosynthetic capacity ("phsyn" in the figure, Chlorophyll Index) (Supplementary File S2, see [36] for greater detail). The box "Landform (hummock)" indicates the effect of the hummock landform relative to the hollow, since that factor is not intrinsically ordered as are the other factors in the SEM. The green arrows indicate the dominant pathway in the hollow landform, while the brown arrows represent the dominant pathway in the hummock landform. (+) symbols represent a positive effect, (−) symbols a negative effect. Non-significant pathways have been excluded for clarity. Photos to the left represent the response to drying of Hollow landforms, while photos to the right represent the response to drying of the Hummock landform. SEM statistical output is presented in Table 1.

**Table 1.** Statistics for structural equation model presented in Figure 4. Significant effects are highlighted in bold.

| Response | Factors | Standardized Effect | Z-Value | P > |z|) |
|---|---|---|---|---|
| Capitulum density | **Water table** | **0.275** | **2.673** | **0.008** |
| | Rainfall | 0.036 | 0.354 | 0.723 |
| | **Landform** | **0.513** | **4.994** | **0** |
| Surface height | **Water table** | **0.67** | **7.279** | **0** |
| | Rainfall | −0.053 | −0.576 | 0.565 |
| | Landform | −0.113 | −1.225 | 0.221 |
| Wetness index (WI) | **Water table** | **−0.653** | **−5.939** | **0** |
| | **Rainfall** | **0.193** | **2.596** | **0.009** |
| | **Capitulum density** | **0.257** | **2.727** | **0.006** |
| | Surface height | −0.174 | −1.652 | 0.098 |
| | **Landform** | **−0.387** | **−4.411** | **0** |
| Normalized difference vegetation index (NDVI) | Water table | −0.013 | −0.159 | 0.874 |
| | Rainfall | −0.027 | −0.557 | 0.577 |
| | **WI** | **0.682** | **8.809** | **0** |
| | **Capitulum density** | **0.118** | **1.933** | **0.053** |
| | **Surface height** | **−0.307** | **−4.656** | **0** |
| | **Landform** | **0.295** | **4.783** | **0** |
| Gross primary productivity (GPP) | WI | −0.189 | −0.846 | 0.398 |
| | **NDVI** | **0.709** | **3.399** | **0.001** |
| | Landform | −0.021 | −0.162 | 0.871 |
| | Water table | −0.181 | −1.335 | 0.182 |
| | Rainfall | −0.094 | −0.976 | 0.329 |

### 4.2. Mire-Breathing

We investigated changes in peat height and capitula density throughout the study in each sphagnocosm, and witnessed not only the standard subsidence "mire-breathing" we had expected [25,41] but also an inwardly compressive "mire-breathing". We found that hollow mosses tended to track the water table downwards, exhibiting a subsidence mire-breathing strategy which has been well documented [25,42]. However, we also found that hummocks exhibited a compressive mire-breathing technique wherein capitula density increased as the mosses compressed into each other (Figure 3). This compression was associated with sustained GPP in response to lower water tables in the hummocks but not hollows.

There are many studies that mention mire-breathing as the swelling and subsidence of peat moss in response to fluctuating water tables [25,42,43]. However, to our knowledge there is very little information on the compressive mire-breathing response that we observed and found to act as a stabilizing factor for GPP.

Some studies have questioned whether "mire-breathing" is a passive reaction simply associated with lowered water tables, or an adaptive mechanism that has evolved over time [25]. Although our study did not investigate evolutionary relationships in *Sphagnum* traits, we found that hummocks exhibited a different, more successful "mire-breathing" strategy than hollow species, arguing that mire-breathing may not be passive, but rather an evolutionary species-specific response. Hummock species did not exhibit vertical subsidence as expected, yet were able to more successfully regulate GPP than the hollow species that did, suggesting a species-specific beneficial trait that may have been preserved through natural selection. Previous studies have concluded that traits related to litter decomposition rates and the niche descriptor height-above-water table are phylogenetically conserved [18]. Therefore, it may not be too far of a stretch to question whether the "mire-breathing" response of hummock species is also phylogenetically conserved.

Piatowski et al. (2021) [18] mapped several species of hummock and hollow forming *Sphagnum* along a height-above-water table decay constant gradient. The two dominant species in our study—*Sphagnum fuscum* (hummock) and *Sphagnum fallax* (hollow)—were located on opposite ends of the gradient as *Sphagnum fuscum* was located high above the water table and had low decomposability while *Sphagnum fallax* was located closer to the water table and had a much higher decomposability. We already know that hummocks form higher above the water table than hollows in part due to slower litter decomposition [18,44], but it is possible that slow decomposition is also linked to greater stability under water stress, and could possibly be a contributing factor behind the inwardly compressive "mire-breathing" that we witnessed in hummocks. As the litter layer in hummocks accumulates, capillary action is most efficient as the live moss pushes closer together, rather than subside towards the water table, which is impeded by the thick litter layer (Figure 3). Whereas in hollows, a thinner litter layer may allow the live moss to simply subside with the water table as we witnessed (Figure 3). Our findings, when grouped with the Piatowski et al. (2021) gradient, suggest that *Sphagnum* species on the far end of the gradient that grow higher above the water table and decompose more slowly, may be more prone to compressive "mire-breathing" and therefore more resilient to water stress than species on the opposite end of the gradient. This supports the idea that evolution in *Sphagnum* dominated peatlands may be favoring slower resource acquisition.

### 4.3. Implications for Peatland Ecosystem Carbon and Water Functions

It is widely debated whether *Sphagnum*-dominated peatlands will transition to carbon sources as *Sphagnum* undergoes water stress as a result of climate change, or if *Sphagnum* may become more productive with longer growing seasons [19,28,29,45–48]. Despite the possibility of longer growing seasons as the climate changes, several studies have found that water is the constraining factor in *Sphagnum* productivity [13,45,49]. In agreement, our study found that *Sphagnum* productivity will be limited when water is limited. Our study supports the prediction that *Sphagnum* dominated peatlands are likely to transition to carbon sources as the climate continues to change and droughts become more frequent or intense. However, our findings also provide hope that peatlands—and specifically hummock forming species of *Sphagnum* moss—are resilient to some extent.

This study draws direct connections between *Sphagnum* traits and peatland hydrology and carbon cycling. Although it is no groundbreaking discovery that hummock mosses are able to better tolerate water stress than hollows [11,19,50], in this study we identified differences in *Sphagnum* landform types, observing a new undocumented form of "mire-breathing", and identified which specific moss traits may enhance the resistance of carbon cycles under water stress. Taken together, this information improves our understanding of how peatlands will respond to altered hydrology.

**Supplementary Materials:** The following supporting information can be downloaded at: https://www.mdpi.com/article/10.3390/w14203239/s1. Figure S1. Time series of change in peat height and $CO_2$ fluxes for the 8 different treatment levels. Please note that the time series for the change in peat height begins 5 days before the time series of $CO_2$ fluxes because the watering and water table treatments were initiated shortly before the $CO_2$ flux measurements. The lines represent the mean for each treatment, data points and error bars are not shown because the plot would not be readable. Significant differences are discussed in the main text. Figure S2. Spectral water content (Wetness Index) and photosynthetic capacity (NDVI) determined at the end of the study using a handheld spectroradiometer. The wetness index was significantly reduced water tables in Hollows, and by both reduced water tables and the interaction of reduced water tables and drought in hummocks, while the chlorophyll index was lowered by reduced water tables in hollows, but only by the combination of reduced water tables and drought in hummocks (see Table S1). Table S1. Linear mixed effects models evaluating water table (WT), rainfall, landform (hummock or hollow) and their interactive effects on two hyperspectral indexes of peat surface moisture and plant/Sphagnum moisture stress. See [36] for details.

**Author Contributions:** Conceptualization, A.O., C.T., E.S.K.; methodology, A.O., C.T., E.S.K.; formal analysis, A.O., C.T.; writing—original draft preparation, A.O., C.T.; writing—review and editing, A.O., C.T., E.S.K.; funding acquisition, A.O., E.S.K. All authors have read and agreed to the published version of the manuscript.

**Funding:** This research was funded by the USDA Forest Service Northern Research Station Climate Change Program, the National Science Foundation (DEB-1146149), and the Michigan Technological University College of Forest Resources and Environmental Sciences.

**Data Availability Statement:** Data will be made available using Github prior to publication of this manuscript.

**Acknowledgments:** We would additionally like to thank the NRS Peatland Reading Group for providing helpful comments and suggestions on an early draft of this manuscript.

**Conflicts of Interest:** The authors declare no conflict of interest.

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
