# Peer review of "Fresh Air for the Mire-Breathing Hypothesis: Sphagnum Moss and Peat Structure Regulate the Response of CO2 Exchange to Altered Hydrology in a Northern Peatland Ecosystem"

_water, doi:10.3390/w14203239_

Round 1

Reviewer 1 Report

General Comments:

This is a well-written paper of a good length that is easy to follow. The authors showed that mire-breathing works differently for hummocks and hollows, which is interesting and important for predicting how a peatland will be impacted by climate change. I think this was a valuable study and I like this manuscript. My comments are intended to help improve the reader’s understanding and interpretation of the research.

Although it is clear that a lot of data were collected, the manuscript feels like a companion paper to other studies, and is somewhat lacking in detail and data. The results are presented as lumped/averaged datasets and generalized concepts, and some results the reader might expect to see were not reported. The Results section could use more data presentation to allow the reader to see the details for themselves. For example, a time series of water table and surface elevation for an average hummock and hollow in response to individual watering events would help illustrate how the sphagnocosms were responding to the rainfall in real time. If these details are reported elsewhere, direct the reader to those papers.

Specific Comments:

Section 2.1: There is no description of how water table depth was measured, or how often it was measured. Even if the details are covered off in Povtin et al. 2015, there should be a brief explanation in Methods as well so the reader doesn’t have to unnecessarily locate the cited article. Furthermore, did the water table ever drop below the 25 cm drain mark due to drought conditions? If so, for how long and did this affect the results? Do you think that deeper cores (e.g. 50 cm) might have produced different or more pronounced results? A time series of the water table in relation to the drain levels would help clear this up for the reader.

L 34: I think “drops in water table depths” should actually be written “drops in water table” or “increases in water table depths”.

L 47: I don’t believe that hummocks and hollows are typically defined by the water table position. Rydin and Jeglum (2013) define hummock as “peatland vegetation raised 20-50 cm above the lowest surface level, characterized by drier-occurring mosses, lichens and dwarf shrubs”. They define hollow as “the lower feature (depression) in peatlands with an alternating microtopography, a hummock-hollow pattern”.

L 56-58: Since a key finding of this study relates to how hummocks and hollows react differently to changes in depth to water table, there should be a brief explanation here of the “classic” mechanics of mire breathing. You could then link your results back to this description.

L 77: This title doesn’t reflect the hydrological methods.

L 78: Some studies have suggested that mire breathing can take place within the entire peat column, or at least more than just the acrotelm. See Kellner and Halldin 2002 and Waddington et al. 2010. Since the depth of your sphagnocosms was only 30 cm, are there limitations to the applicability of these results to actual field conditions? Please discuss. 

Another consideration is that hummocks in some bogs can be very tall (e.g. more than 1 m above surrounding hollows). Could the height of your sampled hummocks have influenced your results because some primarily consisted of undecomposed Sphagnum (i.e. tall hummocks) while others were more peaty (i.e. short hummocks), or were all the sampled hummocks less than 30 cm tall and fairly consistent in their composition? Providing these details would help answer this question for the reader.

L 79: Can you provide a little more information about this peatland? Typical vegetation or structure? Level of disturbance, if any? Perhaps an aerial or ground photograph? Or can you refer to another paper that contains these details?

L 86: There is no Potvin et al. 2015 in the References. Do you mean 2014?

L 93: There is no reference to Figure 1 in the text. This would be a good place.

L 99: How many holes were drilled at each depth? What was the hole diameter?

L 105: Unclear how 16.3 cm relates to the 200 mL and 400 mL noted in Figure 1. I would report rainfall in mm, not cm.

L 115: Should this be section 2.2? It isn’t numbered like the other headings.

L 142-143: Are the Sphagnum species data presented in another paper? If so, cite here. If not, the species data are worth mentioning, as species composition could affect the results (i.e. more than just hummocks = S. fuscum and hollows = S. fallax).

L 161: Be consistent with terminology. I see “Sphagnum type” here (also L 68), but “landform” later on. “Landform” or “landform type” is probably better when referring to hummocks and hollows, as Sphagnum “type” implies species or Sphagnum sections.

L 160-173: There is no reference to Figure 2 in the Results. This paragraph would be a good place.

L 171: What is the purpose of the ?? at the end of this sentence?

L 200: “Water table depth was found to have the greatest effect on peat surface height”. It would be helpful to include a time series showing how peat height and capitulum density react over time to changes in water table depth. Watering events could also be included as a series in the same graph to show how water table reacted to “rainfall”. The peat surface and capitulum reactions are an important finding and deserve a detailed figure.

L 203: Is 3.57 cm (and 1.18 cm) an average or the maximum for a group of samples? Why not report the detailed peat height data somewhere, in addition to the amalgamated data presented in Figure 3? Readers would be interested to compare the mire breathing values to their own research sites.

L 212: The term “compression” here relates to changes in capitula density. This can be easily confused with the compression involved with mire breathing, i.e. vertical compression. Perhaps you can avoid this confusion by using something like “vertical compression” or “surface movement” for the more commonly understood mechanism of mire breathing and “capitula compression” or “inward compression” for the horizontal compression observed in hummock Sphagna.

L 271 and L 292: Consider whether it is necessary to mention “slower resource acquisition” in both locations.

L 296: There appears to be a typo or incomplete sentence: “seasons ga”.

Figure 1: This figure is acceptable but it could be improved to include more graphics relating to the experimental conditions. The high/low WT part is good, because it shows the drains, but why not also show the Sphagnum top at 5 cm above the top drain level (i.e. pre-experiment condition)? The starbursts could be turned into more useful images, like a diagram of a hummock and hollow. Try to make the figure more intuitive so the reader doesn’t have to refer to the caption.

Figure 3: I like the photographic part of this figure. I was disappointed to read that the physical changes depicted here were manually manipulated for illustrative purposes, as it would have been nice to see the actual experimental changes in physical form, but I did appreciate that the manipulation was acknowledged by the authors. Furthermore, if the red arrows actually represent the “mean decrease in peat height or increased density”, include the mean values with the arrows. What do the dashed and solid arrows indicate in “e”? Moving from high to low water table? The graph axes are missing units.

Figure 4: Is there a way to make this model more visually appealing and easier to follow? Maybe use diagrams of the processes to accompany the words. Try to make it more intuitive to follow the processes, like the left side of Figure 3. Improving the visual appeal and clarity of this model could make it much more useful and it would be referenced more often; these findings deserve to be shared and discussed in the literature. Also, is the word “hollow” missing from the “landform” box? What is “phsyn”? This figure should relate to the results of this study, not suddenly bring in results from another study that wasn’t described here (Tucker et al.).

Table 2: There is no reference to Table 2 in the Results or Discussion. Why not, and how do these data relate to those presented in lines 200-203? Why not have three decimal places for all values in the P>z column?

References

Kellner, E. and S. Halldin. 2002. Water budget and surface-layer water storage in a Sphagnum bog in central Sweden. Hydrological Processes 16: 87-103. DOI 10.1002/hyp.286

Rydin and Jeglum. 2013. The Biology of Peatlands, 2nd edition.

Waddington et al. 2010. Differential peat deformation, compressibility, and water storage between peatland microforms: implications for ecosystem function and development. Water Resources Research vol. 46.

Author Response

Reviewer 1

General Comments:

This is a well-written paper of a good length that is easy to follow. The authors showed that mire-breathing works differently for hummocks and hollows, which is interesting and important for predicting how a peatland will be impacted by climate change. I think this was a valuable study and I like this manuscript. My comments are intended to help improve the reader’s understanding and interpretation of the research.

We thank this reviewer for the compliment and the extremely thoughtful and constructive comments! 

Although it is clear that a lot of data were collected, the manuscript feels like a companion paper to other studies, and is somewhat lacking in detail and data. The results are presented as lumped/averaged datasets and generalized concepts, and some results the reader might expect to see were not reported. The Results section could use more data presentation to allow the reader to see the details for themselves. For example, a time series of water table and surface elevation for an average hummock and hollow in response to individual watering events would help illustrate how the sphagnocosms were responding to the rainfall in real time. If these details are reported elsewhere, direct the reader to those papers.

Thanks for pointing that out. We now provide time series of the mean GPP from each treatment, along with a time series of the change in peat heigh and the watering events.

Specific Comments:

Section 2.1: There is no description of how water table depth was measured, or how often it was measured. Even if the details are covered off in Povtin et al. 2015, there should be a brief explanation in Methods as well so the reader doesn’t have to unnecessarily locate the cited article. Furthermore, did the water table ever drop below the 25 cm drain mark due to drought conditions? If so, for how long and did this affect the results? Do you think that deeper cores (e.g. 50 cm) might have produced different or more pronounced results? A time series of the water table in relation to the drain levels would help clear this up for the reader.

Thanks for pointing out the lack of clarity here. This experiment did not have a comparable water table methodology as Potvin et al. (2015), which was cited in this methods section with regard to rainfall chemistry. We did not measure water tables per se, we allowed water tables to equilibrate with the drainage holes and verified that the water tables were at the hole level by probing the drainage holes. we have added the following text: “Water tables remained equilibrated with the hole depth throughout the experiment, as verified by occasionally probing the water level at the drainage hole.”

L 34: I think “drops in water table depths” should actually be written “drops in water table” or “increases in water table depths”.

We have changed this to “drops in water table position” as suggested. 

L 47: I don’t believe that hummocks and hollows are typically defined by the water table position. Rydin and Jeglum (2013) define hummock as “peatland vegetation raised 20-50 cm above the lowest surface level, characterized by drier-occurring mosses, lichens and dwarf shrubs”. They define hollow as “the lower feature (depression) in peatlands with an alternating microtopography, a hummock-hollow pattern”.

Thanks for catching this and for providing the clarification.  We have edited this to read, “…defined primarily by their relative heights with an alternating hummock-hollow microtopographical pattern [Rydin and Jeglum, 2006]”.

L 56-58: Since a key finding of this study relates to how hummocks and hollows react differently to changes in depth to water table, there should be a brief explanation here of the “classic” mechanics of mire breathing. You could then link your results back to this description.

We appreciate this comment and have added in this text, “In the traditional model, the collective peat surface expands as the water table position rises, raising the bog surface and increasing water storage. With a decline in water table position the peat loses volume, and the peat surface drops, which maintains close contact between Sphagnum mosses and the water table—thereby maintaining their productivity.”.

L 77: This title doesn’t reflect the hydrological methods.

Thanks for catching that.  We’ve amended it to read, “2.1. Peat harvest and water table treatments”

L 78: Some studies have suggested that mire breathing can take place within the entire peat column, or at least more than just the acrotelm. See Kellner and Halldin 2002 and Waddington et al. 2010. Since the depth of your sphagnocosms was only 30 cm, are there limitations to the applicability of these results to actual field conditions? Please discuss. 

Another consideration is that hummocks in some bogs can be very tall (e.g. more than 1 m above surrounding hollows). Could the height of your sampled hummocks have influenced your results because some primarily consisted of undecomposed Sphagnum (i.e. tall hummocks) while others were more peaty (i.e. short hummocks), or were all the sampled hummocks less than 30 cm tall and fairly consistent in their composition? Providing these details would help answer this question for the reader.

We really appreciate this comment and the opportunity to put this work in a broader context. We feel that the upper 30 cm captured in this study is within the pocket of these prior studies.  Please see new text, “The full dynamic range of “mire breathing” for hummock/hollow microtopography can range between 20-60 cm depth [Waddington et al., 2010]), with the majority of changes in water storativity occurring in the top 40 cm [Kellner and Halldin, 2002]. As such, the top 30 cm measured in this study is likely reflective of this range but does not necessarily capture dynamics occurring within denser peat.”.

L 79: Can you provide a little more information about this peatland? Typical vegetation or structure? Level of disturbance, if any? Perhaps an aerial or ground photograph? Or can you refer to another paper that contains these details?\

This is a great point, and necessary to put this study in a broader context. The study area has been described in detail, and we now refer the reader to that study.  Please see new text, “Thirty-two cores of peat (~30 cm depth x 20 cm diameter) were collected from a Sphagnum-dominated oligotrophic peatland in Nestoria, MI (46°34′22.66″ N, 088°16′44.85″ W) previously described by Meingast et al., [2014]. Briefly, the over story consisted of Larix laricina and Picea mariana and the understory consisted of mixed sedge and ericoid shrubs.”.

L 86: There is no Potvin et al. 2015 in the References. Do you mean 2014?

Thanks- good catch.  The correct publication year is 2015 (changed). 

L 93: There is no reference to Figure 1 in the text. This would be a good place.

Thank you- we agree. 

L 99: How many holes were drilled at each depth? What was the hole diameter?

We have amended the text in this section as follows: “One drainage hole (3/8” diameter) was drilled in each sphagnocosm 5 cm below the average surface height for that sample, and for the low water table treatments an additional hole was drilled at 25cm below the average surface height. Water was allowed to drain freely from each hole through a perforated tube inserted into the drainage hole to ensure drainage across the peat profile and past the plastic liner. Water tables remained equilibrated with the hole depth throughout the experiment, as verified by occasionally probing the water level at the drainage hole.”

L 105: Unclear how 16.3 cm relates to the 200 mL and 400 mL noted in Figure 1. I would report rainfall in mm, not cm.

Thanks for pointing out the lack of clarity. We have amended the text to read as follows: “Half of the sphagnocosms received a rainfall treatment representing an average monthly summer rainfall of 163 mm while half received a droughted rainfall reduced by ~50% from the average: converting the average rainfall this to units of mL/bin and then dividing by the number of rainfall events resulted in a volume of 400 ml/rain event in the “average” rainfall treatment, and 200 ml/rain event in the “droughted” rain event. Sphagnocosms were watered 12 times over the 25 day duration with artificial rainwater (See Potvin et al. 2015 for rainfall chemistry).”

L 115: Should this be section 2.2? It isn’t numbered like the other headings.

Thanks and good catch.  This has been changed.

L 142-143: Are the Sphagnum species data presented in another paper? If so, cite here. If not, the species data are worth mentioning, as species composition could affect the results (i.e. more than just hummocks = S. fuscum and hollows = S. fallax).

Thank you again.  We have added, “S. magellanicum and S. angustifolium were also present [Meingast et al., 2014].”.

L 161: Be consistent with terminology. I see “Sphagnum type” here (also L 68), but “landform” later on. “Landform” or “landform type” is probably better when referring to hummocks and hollows, as Sphagnum “type” implies species or Sphagnum sections.

Yes, agreed.  We have changed throughout. 

L 160-173: There is no reference to Figure 2 in the Results. This paragraph would be a good place.

Thank you.  Added.

L 171: What is the purpose of the ?? at the end of this sentence?

This was an editing error and has been removed.  Thank you. 

L 200: “Water table depth was found to have the greatest effect on peat surface height”. It would be helpful to include a time series showing how peat height and capitulum density react over time to changes in water table depth. Watering events could also be included as a series in the same graph to show how water table reacted to “rainfall”. The peat surface and capitulum reactions are an important finding and deserve a detailed figure.

Water table was not influenced by individual watering events per se, since that water table remained stable at the level of the drainage holes. We did not measure peat surface height and capitulum density frequently enough to observe responses to individual watering events

L 203: Is 3.57 cm (and 1.18 cm) an average or the maximum for a group of samples? Why not report the detailed peat height data somewhere, in addition to the amalgamated data presented in Figure 3? Readers would be interested to compare the mire breathing values to their own research sites.

The values are the mean differences rather than the maximum. We now present the time series of changes in peat height, along with the time series of COs fluxes in Supplemental 1.

L 212: The term “compression” here relates to changes in capitula density. This can be easily confused with the compression involved with mire breathing, i.e. vertical compression. Perhaps you can avoid this confusion by using something like “vertical compression” or “surface movement” for the more commonly understood mechanism of mire breathing and “capitula compression” or “inward compression” for the horizontal compression observed in hummock Sphagna.

This is a good point.  We have clarified that this is “inward compression” here as suggested. 

L 271 and L 292: Consider whether it is necessary to mention “slower resource acquisition” in both locations.

We agree that these sentences somewhat contradict each other, and therefor have removed the first mention at line 271 (now line near 284). 

L 296: There appears to be a typo or incomplete sentence: “seasons ga”.

Thank you again.  This has been corrected. 

Figure 1: This figure is acceptable but it could be improved to include more graphics relating to the experimental conditions. The high/low WT part is good, because it shows the drains, but why not also show the Sphagnum top at 5 cm above the top drain level (i.e. pre-experiment condition)? The starbursts could be turned into more useful images, like a diagram of a hummock and hollow. Try to make the figure more intuitive so the reader doesn’t have to refer to the caption.

Thank you for this suggestion.  We have altered this to be more illustrative. 

Figure 3: I like the photographic part of this figure. I was disappointed to read that the physical changes depicted here were manually manipulated for illustrative purposes, as it would have been nice to see the actual experimental changes in physical form, but I did appreciate that the manipulation was acknowledged by the authors. Furthermore, if the red arrows actually represent the “mean decrease in peat height or increased density”, include the mean values with the arrows. What do the dashed and solid arrows indicate in “e”? Moving from high to low water table? The graph axes are missing units.

We appreciate this comment and note that these photographs are not actually artificial.  We have clarified this. What we were trying to convey was that in cutting a cross-section of the cylinder monoliths, the actual physical structure was degraded.  The peat cross-sections were therefore re-formed to the measured dimensions taken in-situ prior to harvesting and cutting. Units are now included.

Figure 4: Is there a way to make this model more visually appealing and easier to follow? Maybe use diagrams of the processes to accompany the words. Try to make it more intuitive to follow the processes, like the left side of Figure 3. Improving the visual appeal and clarity of this model could make it much more useful and it would be referenced more often; these findings deserve to be shared and discussed in the literature. Also, is the word “hollow” missing from the “landform” box? What is “phsyn”? This figure should relate to the results of this study, not suddenly bring in results from another study that wasn’t described here. (Tucker et al.).

Thanks for pointing out that there was a lack of clarity in the description of the figure. We have updated the caption to read “Structural equation model of Sphagnum C uptake (GPP) as a function of the treatments, changes in physical structure, and spectral indicators of Sphagnum spectral water content (Wetness Index) and spectral photosynthetic capacity (“phsyn” in the figure, Chlorophyll Index) (Supplement 1, see Tucker et al. 2022 for greater detail). The box “Landform (hummock)” indicates the effect of the hummock landform relative to the hollow, since that factor is not intrinsically ordered as are the other factors in the SEM. The green arrows indicate the dominant pathway in the hollow landform, while the brown arrows represent the dominant pathway in the hummock landform. (+) symbols represent a positive effect, (-) symbols a negative effect. Non-significant pathways have been excluded for clarity. SEM statistical output is presented in Table 1.” We do present the methodology for the spectral data in the methods section of the present paper, and now provide a supplemental section with a figure reproduced from the paper referenced herein.

We have further amended this figure to include photographs of the mesocosm surfaces of a high-average and low-drought water table treatment for each landform, which we believe visually illustrates the pattern observed in the structural equation model nicely.

Table 2: There is no reference to Table 2 in the Results or Discussion. Why not, and how do these data relate to those presented in lines 200-203? Why not have three decimal places for all values in the P>z column?

Thank you for catching this.  In fact, there is only one table, which is mentioned as a companion to Figure 4.  We have now mentioned this in the discussion and in the legend to Figure 4. 

References

Kellner, E. and S. Halldin. 2002. Water budget and surface-layer water storage in a Sphagnum bog in central Sweden. Hydrological Processes 16: 87-103. DOI 10.1002/hyp.286

Rydin and Jeglum. 2013. The Biology of Peatlands, 2nd edition.

Waddington et al. 2010. Differential peat deformation, compressibility, and water storage between peatland microforms: implications for ecosystem function and development. Water Resources Research vol. 46.

Thank you.  These have all been included (see above).

Reviewer 2 Report

The authors investigated the response of CO2 exchange to altered hydrology in a northern peatland ecosystem. Their results are potentially interesting and useful, but I may have the following major and minor comments. The writing is a major concern.

1. To make your results comparable, more details on Materials and methods are required for the further evaluation. For example, how did you select these samples?

2. You need to check data normality before ANOVA analysis. Repeated ANOVA analysis should be used in your study.

3. There is quite large room to improve the writing. Not only for the writing itself but also for the writing logic. The research questions and hypotheses are not clear.  See several relevant studies, Chen et al., 2017, https://doi.org/10.1007/s10021-016-0035-6; Wang et al., 2021, https://doi.org/10.1016/j.agrformet.2021.108388

4. Too many unnecessary abbreviations are preventing the reading. I have to remember a lot abbreviations when reading your manuscript.

5. Line 19-25.  \this is an almighty conclusion sentence. You need to rewrite this sentence for a more informative conclusion and implication sentence.

6. The main conclusions and key implications are not clear enough from your discussion. The potential mechanisms need to be discussed. What are the important biotic and abiotic factors affecting the responses? What are the implications?

7. Some in-depth data analyses are required to explore the underlying mechanisms. Will you compare the relative importance of these variables?

8. the result section is messy. I cannot follow.

Author Response

Reviewer 2

The authors investigated the response of CO2 exchange to altered hydrology in a northern peatland ecosystem. Their results are potentially interesting and useful, but I may have the following major and minor comments. The writing is a major concern.

We thank the reviewer for the constructive comments and have addressed them all, below.

  1. To make your results comparable, more details on Materials and methods are required for the further evaluation. For example, how did you select these samples?

We appreciate this, and direct the reviewer to revised text specifying the specific microforms, “Cores were collected from two peatland landforms: elevated mounds (hummocks) and lower, flat microforms (hollows/lawns, hereafter ‘hollows’) in a relatively open, homogeneous area within the peatland.”.

  1. You need to check data normality before ANOVA analysis. Repeated ANOVA analysis should be used in your study.

Thank you.  Data did not violate model assumptions.

  1. There is quite large room to improve the writing. Not only for the writing itself but also for the writing logic. The research questions and hypotheses are not clear. See several relevant studies, Chen et al., 2017, https://doi.org/10.1007/s10021-016-0035-6; Wang et al., 2021, https://doi.org/10.1016/j.agrformet.2021.108388

We appreciate this comment and have added text to put this study in the broader context of related work.  See comments addressing Reviewer 1, “In the traditional model, the collective peat surface expands as the water table position rises, raising the bog surface and increasing water storage. With a decline in water table position the peat loses volume, and the peat surface drops, which maintains close contact between Sphagnum mosses and the water table—thereby maintaining their productivity. While prior work has shown there to be different water transport and storage characteristics among different peatland microforms [Kellner and Halldin, 2002], it is unknown whether the contrasting Sphagnum landforms (i.e. hummocks and hollows) exhibit similar mechanisms of mire breathing and how that affects their ability to maintain productivity under drier conditions.”.

Further, we appreciate the recommendation to further the context of this work with relevant literature, and have included numerous relevant citations as per Reviewer 1.

  1. Too many unnecessary abbreviations are preventing the reading. I have to remember a lot abbreviations when reading your manuscript.

We regret the GPP and NEE are conventions for this field, and should be preserved. The remote sensing acronyms are only used for context in the discussion. 

  1. Line 19-25. \this is an almighty conclusion sentence. You need to rewrite this sentence for a more informative conclusion and implication sentence.

We appreciate this, and have amended this sentence to read, “Lower peat height was linearly associated with reduced gross primary production (GPP) in response to lowered water tables, indicating that peat subsidence did not significantly enhance the resistance of GPP to drought. Conversely, Sphagnum peat compression was found to stabilize GPP, indicating that this mechanism of resilience may transmit across the landscape, depending on which Sphagnum landform types are dominant.”

  1. The main conclusions and key implications are not clear enough from your discussion. The potential mechanisms need to be discussed. What are the important biotic and abiotic factors affecting the responses? What are the implications?
  2. Some in-depth data analyses are required to explore the underlying mechanisms. Will you compare the relative importance of these variables?

Please see the revised Figure 4, wherein the key biotic and abiotic factors relate to C balance.  Also, please see responses to reviewer 1, above. 

  1. the result section is messy. I cannot follow.

We appreciate this, and have done a more complete job of orienting the reader to the relevant Figures and Tables in the text; please see recommendations from reviewer 1, above.